# Strategic Assessment of Cyber Security Contenders to the Brazilian Agribusiness in the Beef Sector

**Virgínia de Melo Dantas Trinks** [1,*,†]**, Robson de Oliveira Albuquerque** [1,†] **, Rafael Rabelo Nunes** [1,†] **and Gibran Ayupe Mota** [1,†]

Professional Post-Graduate Program in Electrical Engineering—PPEE, Department of Electrical Engineering, University of Brasília, Distrito Federal, Brasília 70910-900, Brazil
* Correspondence: vividantasnatal@gmail.com
† These authors contributed equally to this work.

**Abstract:** The current international commercial structure places Brazilian Agribusiness in constant conflict to protect its interests before other nations in the global market. Technological innovations are used in all stages from the simplest production tasks, up to the design of negotiation tactics at high-level affairs. This paper has the objective of finding Brazilian contenders in the beef market with cyber capabilities and commercial interest to act in favor of their interests. To construct such a list, a review of the literature on Threat and Cyber Threat Intelligence is presented, followed by a background presentation of how embedded technology is in nowadays agriculture and supply chains in general, and the real necessity for those sectors to be seen as critical infrastructure by governments in general. Also as background information recent cyber attack cases and attacker countries are shown. A Step-by-Step multidisciplinary method is presented that involves the extent of international trade, the interest on specific markets, and the intersection of country cyber capacity index. After applying the method and criteria generated a list of five contender countries. The method may be replicated and/or applied, considering adequate data source assessment and following specifics of each sector.

**Keywords:** agribusiness; cyber security; cyber threat intelligence; threat analysis

## 1. Introduction

Agribusiness is essential in today's global economy for the public and private sectors. It is a complex sector that holds interests from diverse players. Similar to other global players of this sector, Brazil's agribusiness is core for the country's Gross Domestic Product (GDP). In Brazil, agribusiness is responsible for more than 20% of the country's GDP, close to one third of its employment, and almost 40% of its exports. Globally, agribusiness represents 10% of consumer spending. This market holds the interests of players that range from powerful governments through large corporations and small building societies [1].

The cattle beef agribusiness chain reached a value of almost USD 40 billion, which account for 15.98% of the Brazilian agribusiness GDP and around 3.64% of the national GDP. If one takes into account only cattle beef participation in Brazilian GDP, there was an increase from 8.4% to 10% percent in 2020 compared with 2019. This demonstrates the primary function of the sector for the Brazilian economy [2].

At the same time, Brazil has registered an 8% increase in cattle beef exports. Of the total beef produced, 73.93% were destined for the domestic market, and the remaining were destined for exports. Of the total exported, there was an increase of 9.8% in the volume of fresh beef; such an increase was due to the expansion in the volume of meat destined for already consolidated markets and to the rise of destination countries, which went from 154 to 157 countries. It is necessary to emphasize the 127% increase of volume exported from Brazil to China in the period of 2021 [2].

According to the Organisation for Economic Co-operation and Development (OECD), meat exports are concentrated, and the combined share of the three largest meat exporters—Brazil, the European Union, and the United States—is projected to remain stable and account for around 60% of global world meat exports until 2030. Brazil, which is the largest exporter of poultry meat, will become the largest beef exporter with a 22% market share by then. The value of the meat trade is dominated by beef and veal, but, in terms of quantity, the meat trade is increasingly dominated by poultry [3]. All these points express the importance of the beef supply chain and export revenue to Brazilian economy, food supply, and society.

A strategic approach to the risk assessment of the Brazilian beef production chain is necessary to expose threats and vulnerabilities and protect the interest of the Brazilian product in the global market. Impact assessments can be used to calculate the level of risk to these assets, from which appropriate remediation steps should follow [4].

### 1.1. Contributions and Limitations of This Work

This paper proposes a strategic assessment that addresses the stated points with focus on creating an analytical scheme that interfaces economic and political interests against cybernetics attack capabilities and intent to introduce an analytics interface that target both academic and corporate issues through Cyber Threat Intelligence (CTI). Therefore, the objective of this study is a model to assess threat analysis of potential contenders and improve threat mitigation through the diagnosis of potential threat actors and their intent. This work will focus on the identification of the main State contenders against Brazilian beef in the international arena and assessment of their cyber attack capabilities according to previous cyber attacks and other known and studied attacks against similar supply chains.

Finally, as a limitation, this paper is not intended to present a risk assessment nor a risk evaluation model as that would require a specific line of research and the strategic assessment of its own with characteristic scenarios, definitions for risk levels, and mitigation. It is our understanding that is a future step to such a study.

### 1.2. Outline of This Paper

The introduction section provides the context of our work while presenting the main goals and limitations. The second section (Section 2) explores important background concepts concerning Food supply chains the impact of technology in Agribusiness nowadays, including Brazilian Beef Production Chain and its relation to technology. It also presents recent cases of cyber attacks against agribusiness sector and supply chain providers. Finally, it demonstrates related works in the threat intelligence area that are crucial for understanding our research. The third section (Section 3) describes the methodology used to construct a list of contenders to Brazilian beef, considering economic, political and cyber power index on a Step-by-Step approach. The fourth section (Section 4) presents how the methodology was put to work in accordance to the previous section in order to reach the proposed objective. The fifth section (Section 5) exposes the results of the countries that have potential to act in the cyber world against the Brazilian Beef Production Chain, that may support other relevant assessment for intelligence gathering on potential risks, actors, and actions. The conclusion in the sixth section (Section 6) provides an analysis of how the results match contemporary concerns over cyber security even in a traditional sector as agriculture and it presents possibilities of future works related to this issue.

## 2. Background and Related Works

The analysis of two different areas exposes the need to evaluate background knowledge to understand contemporary food supply chain structure, and thus to recognize country regulations on infrastructure definition and specific beef sector studies that have produced the necessary data for the assessment presented in this paper. Also, to appropriately assess Cyber Security issues in the agribusiness sector we present definitions and principles in Related Works from Threat Intelligence (TI) and Cyber Threat Intelligence (CTI). Finally, to adequately assess any findings, it is necessary to understand countries

with Cyber Attack Capabilities and their targets, considering trends in the cyber security sector and previous cases presented here as Related Work.

We conducted a review of works about agribusiness and CTI and we did not manage to find prior study that connects the elements presented here in a way to intersect CTI and interested state on the beef agribusiness specific economic sector.

### 2.1. Contemporary Food Supply Context

Demand for food is growing at the same time the supply faces constraints in land and farming inputs. The world's population is on track to reach USD 9.7 billion by 2050, requiring a corresponding 70% increase in calories available for consumption, even as the cost of the inputs needed to generate those calories is rising. Prediction shows that by 2030, the water supply is likely to fall 40% short of meeting global water needs, and rising energy, labor, and nutrient costs are already pressuring profit margins. About one-quarter of arable land is degraded and needs significant restoration before it can again sustain crops at scale.

Environmental pressures are on the rise, also due to climate change and the economic impact of catastrophic weather events. Increasing social pressures highlights the push for more ethical and sustainable farm practices, such as higher standards for farm-animal welfare and reduced use of chemicals and water [5].

All the issues mentioned create a context prone to the increase in price and the complication of production challenges in the food sector; hence, the use of technology in the area is likely to guide and ease producers adaptation to a new world. Countries need to prepare for the upcoming circumstances surrounding food production, especially large exporters that are economically dependent on their food commodities revenues, such as Brazil. It is necessary to ensure productivity and reduce food scarcity that might cause civil unrest and societal tumult.

After the 9/11 attacks, the USA updated its definition of Critical Infrastructure (CI) to include *"Systems and assets, whether physical or virtual, so vital to the USA that the incapacity or destruction of such systems and assets would have a debilitating impact on security, national economic security, national public health or safety or any combination of those matters"* [6]. According to Ossevorth et al. [7]:

> *"In this context resilience, which is defined as the resistance of a system to external effects, is required. A field that is indeed part of the critical infrastructure, but which has not been considered as intensively as the energy sector, is food production."*

In the USA, the Cyber security and Infrastructure Security Agency (CISA) [8] understands that, amongst others, the Food and Agriculture Sector is one of the infrastructures that need protection under federal regulation. The regulation also recognizes that each infrastructure sector possesses its unique characteristics and operating models. Finally, it is highlighted which sectors hold dependencies with the Food and Agriculture Sector.

In Brazil, CI was defined by decrees no. 10.569, 2020 [9], and no. 6.703, 2008 [10], as strategic facilities, services and goods whose interruption or destruction will cause a serious social, economic, political, international or national security impact, in particular in the sectors of energy, transport, water and telecommunications. Therefore, the decrees state that those facilities need security measures capable of guaranteeing their integrity and functioning, which means that physical and operational security needs to be known and monitored in order to ensure the provision of those essential services.

Agriculture, food production or protection of commodities or commercial interests are not mentioned in any of those federal regulations, even though Brazilian legislation provides cooperation in protecting national CI, by monitoring threats related to acts of sabotage that might threaten the functioning of those strategic facilities.

In Europe, the OECD classifies six sectors as CI: information and communication technologies, energy, finance, health, transport and water [11]. Food supply appears in a second group of sectors that includes government, chemical industry, or public safety, for about half of the countries. An OECD white paper considers that the list of critical sectors

can evolve over time to address emerging vulnerabilities and evolving risks and that has lead to differences in categorisation across countries.

In Canada, the National Strategy for Critical Infrastructure [12] establishes a collaborative, federal-provincial-territorial and private sector approach built around partnerships, risk management and information sharing and protection. The central idea is that the national strategy may give a coherent and complementary approach to the 10 chosen sectors in order to strengthen resiliency across jurisdictions, food supply is considered one of those sectors.

In Japan, the National Strategy for Critical Infrastructure Protection [13] admits that there is suspicion of the involvement of national governments in targeted attacks aimed at stealing secret information such as trade secrets, and that cyber attacks against Japan involving the participation of foreign governments could occur. Therefore, Japan's regulation affirms that there are also fears of possible future attacks in a global supply chain. In this context, even though food is not nominated as a CI by Japanese authorities, the country is aware of the necessity of protecting basic supply chains.

In Australia, the Trusted Information Sharing Network (TISN) for Critical Infrastructure Resilience was established by the Australian Government in 2003. The TISN provides national level forums for owners and operators of CI to develop strategies and solutions to mitigate risk in the following sectors: Energy, Water, Communications, Banking and Finance, Health, Transport, and Food [11].

### 2.1.1. Impact of Technology in Agribusiness and Strategic Supply Chains

Contemporary agriculture is in the early days of a revolution, at the heart of which lie data and connectivity. Artificial intelligence, analytic, connected sensors, and other emerging technologies could increase yields, improve the efficiency of water usage and other inputs, and build sustainability and resilience across crop cultivation and animal husbandry. With the implementation of connectivity in agriculture, the industry could add on USD 500 billion in value by 2030 [5]. Connectivity infrastructure is expected to cover roughly 80 percent of the world's rural areas, with the exception of Africa, in this context, the key is to develop effective digital tools for the industry, and to foster their adoption [5].

Technological developments bring an infinite horizon of possibilities and uses for beef production, for example, the massive Internet of Things, low-power networks, and cheaper sensors should monitor large herds of livestock, and track the use and performance of remote buildings and large fleets of machinery, which are mission-critical services. Ultralow latency and improved stability of connections will foster confidence to run applications that demand absolute reliability and responsiveness, such as operating autonomous machinery and drones. If LEO satellites attain their potential, they will probably enable even the most remote rural areas of the world to use extensive digitization, which should enhance farming productivity [5].

The unavoidable deployment of 5G networks should impact the sector, once IoT can inherently support a significant number of more connected devices and facilitate industrial adoption and employment of automation systems. Open RAN reduces capital and operational expense levels and improves deployment agility, but it also lacks security focus, as evidenced by various Open RAN alliances [5].

Proper security planning and investments become primordial to conform to those new realities, even to a tradition-related sector such as agribusiness. Recent cases of strategic supply chains workflow being challenged after cybernetic attacks show that modern production of key products is heavily automated, not only for safety reasons.

### 2.1.2. Brazilian Beef Production Chain

The beef production chain starts in the input sector. Then, it passes through the production sectors, where the slaughterhouses transform the raw material into a finished product. Finally, distribution to the retail segment is responsible for the advancement of end product towards the consumer [14]. Aspects related to foreign trade, macroeconomic

evolution, inspection, legislation, product availability, reliability of statistical information, environmental legislation, trace-ability and certification mechanisms, innovation systems, among others, strongly condition the competitiveness in the sector.

It is strongly recommended that livestock farmers use tools that minimize the impact of price volatility in the livestock market on their business for the long run. In the last 20 years, Brazilian beef was able to reduce operational costs due to the increase of technology use. As a result, the amount of not inspected beef produced dropped from 50% percent to less than 22% [2].

Even though specifics of beef production make it difficult to perceive the advances that took place along the production chain, those numbers show that, slowly and steadily, Brazilian beef production is moving towards what is seen internationally as Precision Livestock Farming (PLF) [15], which enables the collection of more precise data.

Technology should supply farmers with more precise data, broader management options, possible productivity increase, better disease control or healthier flock, food safety improvement in general, etc. Total production costs of farms that count with the complete cycle of six levels of technology are much lower than those who do not [2].

The 2021 edition of the OECD-FAO Agricultural Outlook [3] projects the global meat supply to expand over the projection period, reaching 374 Mt by 2030. Herd and flock expansion, especially in the Americas and China, combined with increased per animal productivity (average slaughter weight, improved breeding, and better feed formulations) will support the meat market. This explains the importance of the use of technology in the interest of lowering costs, improving effectiveness, food safety, product availability, and organizations' reliability, and security as a whole.

Malafaia [16] explains that the Brazilian beef cattle supply chain has undergone technological modernisation in its production systems, resulting in better productivity, meat quality and competitiveness. This demonstrates that the Brazilian food sector is central to the world economy and, as such, it is the point of interest of a wide range of actors. It is an international reality that weaponizing CI has become a means to undermine countries capabilities in a contemporary Hybrid Warfare format [17].

### 2.2. Threat Intelligence

According to Chismon and Ruks [18], it is relevant to have a clear differentiation between vulnerability information and threat intelligence to produce relevant intelligence. A vulnerability might exist in a product used by the organisation that does not necessarily have information about a particular threat. Considering traditional intelligence versus today's world of effective and motivated attackers, with country funding and resourcing, it is critical that security principles are valued.

Threat intelligence formation has yet to have an exhaustive format and methodology. Companies, countries, and academia are learning and improving on a day-by-day basis, taking into account contemporary occurrences and fast technological development. Traditional Threat Intelligence is still relevant in the sense that it comes from observation and analysis of contenders and that it [19]:

> *"Must be actionable to meet the needs of current defensive systems that have to deal with and respond to cyber attacks."*

Consequently, trends in country strategies, ambitions, priorities and other high-level information should instruct strategic analysis. That information needs to be coupled with observations of malware or cyber attacks thought to create a picture of cyber activities. High-level sources need to feed this type of information to Threat Intelligence analysts including [18]:

> *"Policy releases by nations or groups of interest, news stories in domestic and foreign press, and news stories in subject-specific press, such as financial papers, or articles published in journals by high-ranking persons in the nation or group of interest, as all of those can be indicators of intent or capability."*

2.2.1. Cyber Threat Intelligence

Even though there is a general awareness of the need for CTI nowadays, it is an undeveloped field that follows the basic principles of traditional Intelligence production cycle and that should consider all details around an effective and multifaceted security system [20]. In this matter, ISO introduced an updated version of the ISO 27001 in 2022, named ISO 27002. One of the most crucial facets of this standard includes threat intelligence and it enables companies to collect and analyze data. CTI in ISO standards aims at protection by increasing awareness of the threats inside or outside of the organization [21].

According to Tounsi's work [4], the most used defense techniques and tools commonly rely on static malware signatures that might leave organizations vulnerable to ever-evolving threats that exploit unknown and zero-day vulnerabilities. This ever-changing scenario requires a new format of threat prevention tools and planning that adapt to the complex nature of new generation threats and work on a more precise aim for threat analysts and tools. The concept of CTI is intertwined with the one of TI in the sense that they constitute evidence-based knowledge representing threats that may inform and support the decision making process. Hence, CTI can be perceived as a process that helps to reduce the gap between advanced attacks and defense mechanisms.

It is relevant to understand the definition of Cyber Security as the protection of information systems (hardware, software and associated infrastructure), the data on them, and the services they provide, from unauthorised access, harm or misuse. This includes harm caused intentionally by the operator of the system, or accidentally, as a result of failing to follow security procedures [22], and thus to fully grasp the importance of CTI and to protect the sector accordingly.

Some analytical frameworks provide structures for thinking about attacks and contenders to allow defenders to take decisive actions faster. For example, the defensive perspective of a kill chain and the Diamond model used to track attack groups over time [4].

With respect to updated cyber security necessities, Agribusiness reality and current CTI production cycles as presented by Borges et al. [23] presented a strategic approach to understand how CTI may assist interested parties to develop long-term cyber security strategies. Thus, intersecting CTI with economic and political components may lead to thorough and updated assessment for the unveiling of potential cyber threats.

Tounsi [4] and Evans [17] provided key definitions on CTI, and how they are currently being used in the kinetic world, through International Relations and warfare. In addition, we were able to grasp how the literature subdivides the issues surrounding those topics and the emerging research studies, trends, and standards that might mitigate those issues.

The work of Shin and Lowry [24] highlighted the reasons why CTI ascended from a growing demand of organizations to understand their enemies and plan accordingly for proactive, preventive, and timely threat detection, with focus on improving 'general readiness' against known or unknown threats. In this sense,

*"CTI represents actionable threat information that is relevant to a specific organization".*

2.3. *Countries with Cyber Attack Capabilities and Their Targets*

The 2021 Threat Landscape Report of the European Union Agency for Cybersecurity (ENISA) selected state-sponsored actors as a category to be highlighted due to its prominence during the reporting period. According to the report state-sponsored threat were observed targeting healthcare, pharmaceutical, and medical research sectors, throughout the COVID-19 crisis. Apparently, the collection of scientific information related to the COVID-19 vaccine was a high priority [25]. The report also recognized that supply chain compromises by state-backed threat actors are not new, and that this type of attack has reached new levels of sophistication and impact since 2021.

The acts might occur for strategic objectives or for personal gain, and with varying levels of national responsibility, which sheds doubt on the definitions of cyberespionage and cybercrime operations.

The main spotted trends in the sector showcase that countries with advanced cyber capabilities are using these to strategically shape global political, military, economic, and ideological power, while middle powers are focusing on initiatives related to regulation, cyber norms, and protection of their critical infrastructure. Cyber operations are aligned with the strategic objectives of states as well as the geopolitical landscape and real-world events.

ENISA also highlighted, among other examples, increased cyber intrusion activities in regions of trade routes, against strategic targets such as governmental organisations, and cyber operations as enablers for large-scale espionage. This movement is not only here to stay, but will be increasingly used for intelligence gathering and critical infrastructure attacks. Thus, state-sponsored groups are expected to conduct operations to weaken, demoralise, and discredit adversarial governments and install media misinformation in order to amplify impact through the exploitation of societal divisions, trust impairment, and society polarisation over issues that are sensitive in certain countries [25].

The Guide to Developing a National Cybersecurity Strategy by the International Telecommunication Union (ITU) [26] stressed the importance of international law enforcement cooperation and formal or informal mechanisms to share information, build trust, and support cross-border cooperation in combating cybercrime and other cyber-enabled crimes. The ITU guide recognized that:

*"To fully realise the potential of technology, states must align their national economic visions with their national security priorities."*

This means that nations should be working on offensive and defensive capabilities to defend themselves from illicit and illegal activities in cyberspace, and to pre-empt incidents before they can cause harm.

In an attempt to understand actors in the sector, a group of researchers at Harvard University came up with the Harvard National Cyber Power Index (NCPI) index [27] that considers that the analysis of cyber power is the product of intent and capability. As a result the top 10 "most comprehensive countries" with the highest level of Intent Ranking by Commercial Objective are as follows [27]:

1. China
2. Iran
3. United Kingdom
4. Japan
5. Switzerland
6. The Netherlands
7. Sweden
8. Australia
9. USA
10. Russia

The NCPI considers Cyber Power as the product of intent and capability, so countries with a high level of those characteristics are among the highest-ranking countries in the Index. These countries have shown both in strategies and in previously-attributed cyber-attacks that they intend to use cyber to achieve policy goals and have the capabilities to achieve this.

The index recognizes countries not normally associated with cyber powers, due to their strong capabilities in certain areas. For example, Sweden is ranked in the top 10 for surveillance, cyber defense, and information control, and Switzerland made the top 10 for cyber defense and commercial gain.

China deserves an explanation of its own: it has been found to use industrial espionage, to incentivize and grow its domestic cyber expertise through research and development, and public-private partnerships, both in a legal an illegal manner.

Finally, it is likely that state-backed threat actors will continue conducting supply chain attacks, especially targeting software, cloud, cloud-hosted development environments, and

managed service providers, that is not to forget that cybercrime threat actors increasingly show the same patterns of behaviour [25].

*2.4. Recent Cases of Vulnerabilities Exploitation*

Considering cyber offense trends, countries with great dependability on world commerce and their exports need to adapt to the contemporary commerce world and introduce themselves to this interconnected war with investment and planning. Cyber threats grow rapidly, promoted by the rise of digitization, this expansion comes with dangers and target amplification. Businesses digitization courses may only be successful if proper cyber security techniques are employed [1]. In this environment, cyber attacks have become more common. Below we present recent cases of cyber attacks against the agribusiness sector, and relevant supply chain providers in chronological order.

### 2.4.1. JBS Attack

On 30 May 2021, newspapers all over the world reported on the case of the Brazilian-based meat company called JBS that had its servers and computer networks attacked, temporarily shutting down some plant operations in Australia, Canada, and the USA. Even though backup servers were not affected, the attack caused delay in transactions with clients and suppliers, and damaged the company's image, and a discussion commenced over possible meat shortages and price rises. Only by the beginning of June was the company able to fully recover and put its global IT Systems back in order.

Crisis management steps were taken to handle the situation: JBS facilities in the American States of Michigan and Iowa were temporarily closed, some Australian facilities operations were suspended and others operated at a limited level. That disruption threatened food supplies and risked higher food prices for consumers.

The White House has said that a criminal organisation "likely based in Russia" was behind the attack. American National Security organizations expressed their concern because it affected the food supply chain, which is fundamental for the health of the nation. As a result, there were political actions towards sanctions against possible threat actors, emergence of new cryptocurrency rules, and negotiations to turn ransom payoff into a crime were evoked.

On 4 June 2021, Russia-linked cyber group REvil announced it was responsible for the JBS attack via an interview to Sergey R3dhunt in Telegram, in which he said the attack targeted Brazilian Operations of JBS initially. On 10 June, JBS announced it had paid USD 11 million in ransom to put an end to the attack, the payment was reportedly made using Bitcoin after plants had come back online.

On 16 June 2021, the American and Russian Presidents held a summit in Geneva, where Cyber security was a significant topic of conversation. The American President clearly stated Cyber security was a vital American interest and stated that "Russian activities that run counter to those interests will be met with a response" in an intimidating discourse. That fractured relations of Russia and the USA.

### 2.4.2. John Deere and Case New Holland

In August 2021, a group of hackers called Sick Codes made a presentation at the DefCon security conference showing how they had used the John Deere platform to make changes to supply networks, equipment reservations and even the contact details of those who received "demo units" from the company.

### 2.4.3. USAHERDS

In March 2022, a China-affiliated threat actor, known as APT41 or Barium, used Log4j and zero-day bugs to breach at least six US state governments networks for over a year. APT41 used a vulnerability in the USAHerds—Animal Health Emergency Reporting Diagnostic System—to penetrate state networks. The software is used by 18 states throughout the USA; all of them are now under scrutiny to understand if their servers could have

been invaded or even hijacked by the hackers. The Barium group has not yet disclosed its objective nor what data they may have been seeking.

## 3. Proposed Methods and Criteria Evaluation

To reach the objective of identifying Brazilian contenders in the beef market with cyber capabilities and commercial interest to act in favor of its interests, we created a methodology which is presented below Step-by-Step (see Figure 1 for visual).

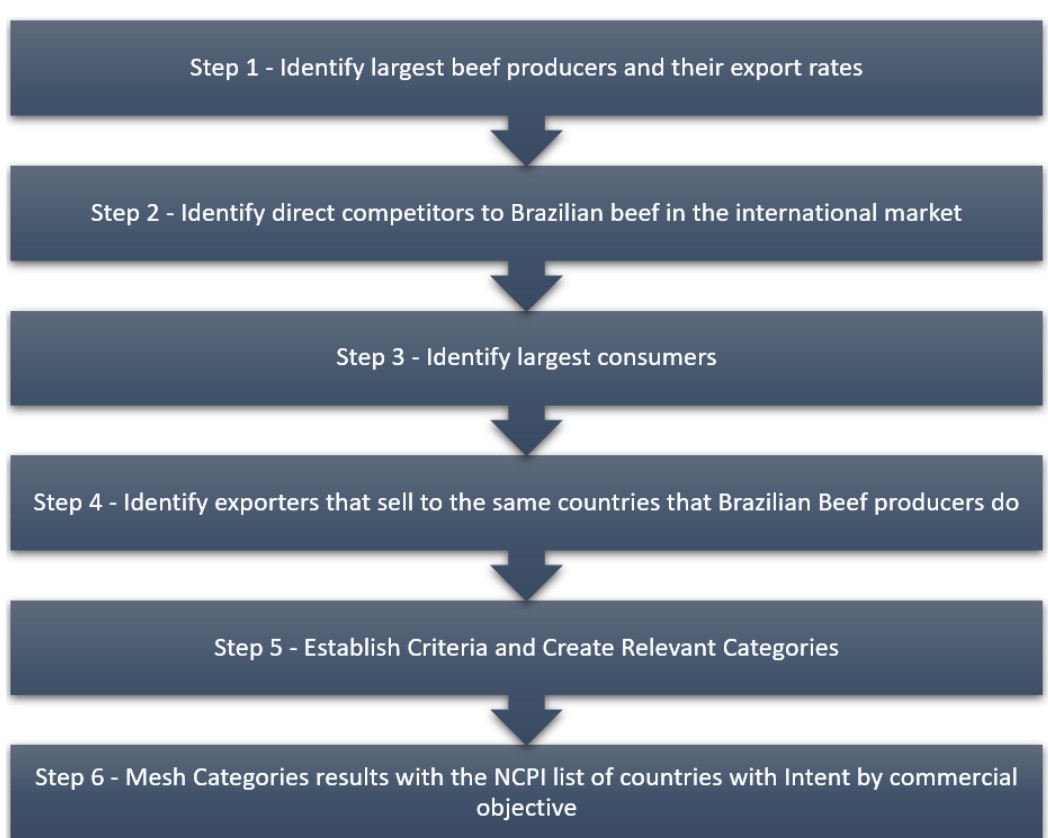

**Figure 1.** Methodology to find Brazilian beef contenders with cyber attack capacities and intent.

### *3.1. Step 1*

First we identified the largest beef producers in the world. We analyzed if their export rate was indeed significant for the whole country state export revenue. The idea is to understand if among big producers that are also large consumers that may not be interested in the export market. The analysis resulted in a list of countries where beef exports have significance for the whole country export revenue, ergo countries that have true interest in the international beef market.

### *3.2. Step 2*

Second, we focused on finding direct and de facto competitors of Brazilian beef in the international market. To reach that end we identified top exporters, then we intersected the top exporters list to the findings of Step 1 to understand if the largest exporters list is congruent to the largest producer one. After, we assessed and compared the lists and we could name which countries are actively involved in the beef world market competition as exporters.

### *3.3. Step 3*

Third, we considered the role biggest buyers play in the market, not only to have a wider understanding of the market but also to make sure what countries rely on Brazilian

and foreign exports to guarantee their product supply. Thus, those countries' interests are presented in terms of price perspective, as well as for the guarantee of access to good product according to food safety and security issues.

### 3.4. Step 4

Fourth, we identified which countries were direct competitors of Brazilian beef in its specific importer markets. For that to occur, we listed main providers of beef to each of the biggest buyers that were found in the third step to reach a list of the countries that export to the same countries that Brazil does.

### 3.5. Step 5

The criteria established in the previous steps lead us to understand the competition and identify contenders according to economic and political interests. It is noteworthy that assessment showed that, to find the contenders in a specific market, a country must consider not only its own exporters but also importers interests and big producers' strategies within the world market. Hence we reached the following five categories of participants in the world market; each category holds a list of countries (see Tables for visual aid):

- Countries that produces the most, result of Step 1 (named 'Producers' in Table 1);
- Countries that exports the most, result of Step 2 (named 'Exporters' in Table 2);
- Countries that compete directly against Brazilian product because they share clients, result of Step 4 (named 'Competitors' in Table 3);
- Countries that imports more from Brazilian product, first result of Step 3 (named 'Importers' in Table 4);
- Countries that consume more beef in the world, second result of Step 3 (named 'Consumers' in Table 5).

**Table 1.** Largest Beef Producers and Export Rate.

| Country | Beef Production (milTEC *) | Export/Production % (mil TEC) |
|---|---|---|
| USA | 12,347.7 | 9.94 |
| Brazil | 10,187 | 26.42 |
| European Union (EU) | 7665.7 | 44.73 |
| Argentina | 3178.5 | 27.06 |
| India | 2474.9 | 31.2 |
| Australia | 2078 | 66.79 |
| Canada | 1304.7 | 36.15 |
| New Zealand | 703 | 84.41 |
| Ireland | 649 | 86.97 |
| Poland | 605.1 | 100.15 |
| Uruguay | 514.5 | 74.19 |
| The Netherlands | 396.3 | 162.41 |

* Thousand Tons of Carcass Equivalent.

### 3.6. Step 6

After establishing a list of contenders from a political and economic perspective, we intersect the results with the NCPI index as mentioned in Section 2.3. We check what countries are mentioned in at least three categories of Step 5 and that are also listed in the NCPI top 10 Index to come up with the result of a list of countries that are agribusiness contenders and that hold cyber capacities and have intent.

**Table 2.** Largest Beef Importers and Consumers.

| Country | Imports (milTEC) | Importers of Brazilian Beef in Percentage |
|---|---|---|
| EU | 5886.7 | 6.24 |
| China | 2223.4 | 50.8 |
| USA | 1459.3 | 41.67 |
| Hong Kong | 619.6 | 60.81 |
| The Netherlands | 472.2 | 32.9 |
| Italy | 424.4 | 9 |
| Egypt | 403.9 | 41.67 |
| Russia | 344.8 | 21.69 |
| Chile | 283.1 | 41.69 |
| Uruguay | 46.7 | 74.96 |

**Table 3.** Competitors within Consumer Markets.

| Country | Beef Export Competitors (Source ITC) |
|---|---|
| EUA | Australia, New Zealand, Nicaragua, Uruguay, Mexico, Canada, Japan |
| China | Australia, New Zealand, USA, Argentina, Uruguay, Brazil |
| EU | Brazil, Australia, USA, India, New Zealand, Local producers |
| Chile | Brazil, Paraguay, USA, Argentina, Uruguay, China |
| Egypt | Brazil, India, Paraguay, Colombia, Australia, New Zealand |
| Russia | Brazil, India, Paraguay, Belarus, Argentina, India, Colombia |

**Table 4.** Criteria comparison of all stakeholders.

| Countries | Producers Step 1 | Exporters Step 2 | Importers Step 3.1 | Consumers Step 3.2 | Competitor Step 4 | Mentions Step 5 |
|---|---|---|---|---|---|---|
| Argentina | X | X | N/A | X | X | 4 |
| Australia | X | X | N/A | X | X | 4 |
| Canada | X | N/A | N/A | X | X | 3 |
| Chile | N/A | N/A | X | X | N/A | 2 |
| China | N/A | N/A | X | X | X | 3 |
| Egypt | N/A | N/A | X | X | N/A | 2 |
| EU | X | X | X | X | N/A | 4 |
| India | X | N/A | N/A | X | X | 3 |
| Ireland | X | X | N/A | N/A | N/A | 2 |
| Japan | N/A | N/A | N/A | X | X | 2 |
| Mexico | N/A | X | N/A | X | X | 3 |
| The Netherlands | X | X | X | N/A | N/A | 3 |
| New Zealand | X | X | N/A | X | X | 4 |
| Poland | X | X | N/A | N/A | N/A | 2 |
| Russia | N/A | N/A | X | X | N/A | 2 |
| Uruguay | X | N/A | X | X | X | 4 |
| USA | X | X | X | X | X | 5 |

**Table 5.** Brazilian Beef Conceivable Contenders, Step 6 of the methodology (Section 3.6).

| Brazilian Conceivable Contenders in the Beef World Export | Number of Categories Mentions | State Ranked on the NCPI |
|---|---|---|
| USA | 5 | 9th |
| Australia | 5 | 8th |
| EU | 5 | N/A |
| New Zealand | 4 | N/A |
| Uruguay | 4 | N/A |
| Argentina | 4 | N/A |
| Mexico | 3 | N/A |
| Canada | 3 | N/A |
| India | 3 | N/A |
| China | 3 | 2nd |
| The Netherlands | 3 | 6th |
| Ireland | 2 | N/A |
| Russia | 2 | 10th |

## 4. Results and Evaluations

In this section, the methodology presented in Section 3 is put in action as the international beef market is analyzed and presented with focus on our objective in order to find list of countries to be assessed and compared to the index previously explained in Section 2.3.

### 4.1. Identification of the Largest Beef Producers and Their Export Rates

The largest beef producers and their export rate demonstrate in what countries beef exports have significance for the whole export revenue [28] in accordance to Step 1 (see Section 3.1). Table 1 demonstrates that the beef market is yet more complex and a deeper discussion of the numbers is required to be able to identify contenders appropriately.

First, it is notable that the USA is the largest producer, and nearly 10% of its production is destined to export. Considering the large amount of production the export rate is enormous for the USA. Also, it is important to point out that other big producers such as Australia, New Zealand, Ireland, and Uruguay have an export rate of production higher than 50%, which indicates their producers are more dependent on exports and world market. It is interesting to point out the cases of the The Netherlands and Poland that are exporting more than they are able to produce, an indicator that they are importing and selling part of their imports, either for a differentiated treatment capacity or due to geopolitical advantages.

### 4.2. Identification of Direct Competitors to Brazilian Beef in the International Market

In light of the information extracted from the production, it was necessary to analyze if the largest exporters are congruent to the largest producer states to see what countries are competing in this market. With that information, we verified that the interested countries in the exporter market are virtually the same as those in the largest producers list [2]. In accordance with Step 2 (see Section 3.2), the analysis of Table 1 lead to the following list of countries potentially interested in the international market: Brazil; Australia; USA; Argentina; India; The Netherlands; Poland; New Zealand; Ireland; and Canada.

### 4.3. Identification of the Largest Consumers

In accordance to Step 3 (see Section 3.3), we assessed which countries are the largest importers in the world market and how much of their intake comes from Brazilian products [28]. Table 2 displays how Brazil represents a large portion of all those markets.

### 4.4. Identification of Exporters That Sell to the Same Countries That Brazilian Beef Producers Do

The next assessment is presented in Table 3 to understand which countries were direct competitors of Brazilian Beef in those importing markets, in accordance to Step 4 (see Section 3.4). This means that the mentioned countries target their export to the same

countries that Brazil does [28]. That assessment lead to the following countries: Australia; New Zealand; Uruguay; Mexico; Canada; Japan; USA; Argentina; India; and China.

### 4.5. Establishing Criteria in Accordance to Relevant Categories

After gathering the results from all previous assessments, we reached five categories of active participants in the world market that needed evaluation. In accordance with Step 5 (see Section 3.5), we intersected the five categories and we were able to spot the players that hold interest in the Brazilian participation in the market. We listed the players that appeared in at least in 3 out of the five categories, and the results lead to the classification shown in Table 4.

### 4.6. Meshing Categories Results with the NCPI Index

After going through the first steps that considered political and economic matters, it was necessary to out intersect those results with a country index that considered CTI principles. It has been shown that the analysis of cyber power is the product of intent and capability [27] for multiple cyber objectives, with a specific aim to provide an Intent Ranking according to commercial objectives and assess the proposed multidisciplinary intersection in accordance with Step 6 (see Section 3.6). The focus of the analysis was on commercial interests in agribusiness markets; thus, we were able to list the main Brazilian Agribusiness World contenders with Cyber Capabilities in the Beef Sector, as presented in Table 5.

### 4.7. Cyber Security Contenders to the Brazilian Agribusiness in the Beef Sector

Note that, for those states that have not received an applicable grade that does not mean they do not have Cybernetics Commercial Objectives and capabilities. It is merely a categorization fact that they were not ranked in the top 10 countries as such.

The EU and Russia are special cases that need clarification. First, the EU group of countries is not listed as one country when it comes to the NCPI ranking system, so, even though it receives 5 mentions as a player of interest in the market, we could not place it in a specific rank grade when it comes to cyber capabilities. That does not mean the EU or its members do not hold or could not act to favor its interests in cyber space.

It is noteworthy that, in terms of intelligence, some EU countries maintain a traditional history of protecting their commercial interest. Finally, when it comes to Russia, recent chapters of world history regarding Ukraine have shown Russia's growing intent of using its cyber capabilities to protect its own interests. Thus, despite the fact that it received only 2 mentions in the market players charter, it is not a contender to be taken lightly.

Finally, we reached a list of five contenders to the Brazilian Beef in the International Market that hold cyber attack capacities; the list includes the USA, Australia, China, The Netherlands and Russia.

## 5. Discussion

In the competitive world presented in Section 2.1.2, it is primordial to guarantee useful production intelligence to subsidize the decision making process and to defend against possible threats as successfully as possible. This context lead us to cross-reference the economic interests of rival states to Brazil on the specific product market according to International Relations principles versus the operational threat intelligence information and capacities of such states. We considered National Economic Intelligence (NEI) principles that embrace State Intelligence for the purpose of economic development. This is consistent with a set of coordinated actions for the search, treatment, dissemination, and protection of information useful to different economic actors, with the effective employment of Intelligence Services for economic purposes [29].

One must consider the principles and structure presented by the concept of NEI as it introduces the need for broad governmental approaches that give the issue the necessary

applicability with a focus on producing quality intelligence and visualization in a timely manner.

Current international relations require [4]

*"organizations looking to have technical threat intelligence are now overwhelmed with a massive amount of threat data, leaving them with the huge challenge of identifying what is actually relevant. Thus, a problem of quantity over quality has been developed."*

In this context, the objective is to produce structured information that will give strategic CTI directives about what states are most likely interested in Brazilian exports of beef, and what their cyber capabilities are in a hypothetical attack scenario according to Section 2.2.1. This may produce knowledge to support threat categorization and defense prioritization for Brazilian Beef Exporters as we propose in the results section of this work.

In regards to countries' capacities, we start with the public knowledge that the United States has done a number of exercises on cybersecurity, but the results of some are highly classified, making it difficult to evaluate the actual risks present on the cyber domain. Other countries, notably Russia and China, were able to recruit cyber volunteers (both internally and in the diaspora) for militia-like attacks. Cyber conflict is a general term that goes from low-level intrusions, through apt ransomware attacks, or even petty crimes to create spam networks, up to high-scale, state-sponsored cyber warfare and influence operations.

The U.S. beef industry competes with Canada, Australia, New Zealand, Brazil, Argentina, and Uruguay for the export market. In this scenario, even though significant trade barriers exist, there are still opportunities as beef consumption rapidly grows and creates room for global expansion of the beef industry [30]. This means that buyers also have an interest and play a role in the market, not only in the interest of price, but also for the guarantee of access to good products according to food safety and security issues, as seen in Section 4.3.

On the other hand, cybernetic cases like Stuxnet, Flame and Duqu cyber campaign against Iran (codenamed Olympic Games) in 2009–2010 and WikiLeaks' release of thousands of diplomatic cables pertaining to the US State Department and its Missions abroad 2010–2011 have clearly exposed state activities on the edge of legality in order to reach its goals, according to Gamero-Garrido [31]. In this context, major global commerce players should be aware of other players' capacities and the possible ways they might act.

The work of Gamero-Garrido [31] demonstrates that, even though cybernetic conflict cases in the last 30 years were diverse in their scope, actors, tools used, and outcome, it is safe to say the majority of those cases fit one of the following categories: espionage, attack or warfare, and public release of secret government information.

Cyberspace introduced a new field of play to International conflict of interests, and organizations are bound to conform to this reality. Traditional actors have increasingly recognized the importance of the domain, and they are investing in strategies to assert themselves. Actions range from proposals of cybernetic global regulation, through expansion of International cooperation, including both public and private actors, and a closer involvement/monitoring of critical private sectors.

All in a context where cyberspace remains unstructured, according to Cardon [32]:

*"Especially when considered in the context of a political map, detailing the physical and sovereign boundaries between nation states. Without physical delineations to define jurisdictions, the established law, authorities, regulations, processes, structure, and concepts applied to the cyber domain are still in flux for both the public and private sectors."*

Thus, deterring, detecting, mitigating, reporting, and monitoring are the set list for defenders. They must act not only according to the sector's needs but also considering CTI specifications (as discussed in Section 2.2.1). Best practices nowadays dictate multidisciplinary teams, and protection planning following strict risk assessment in order to reach security levels that hold all, from supply chain risks to internal attacks, in a joint effort from private and public bodies.

For example, the US Government announced that their intelligence would conduct a 60-day "sprint" exercise focused on battling ransomware and for that they provided USD 25 million in grants to state and local cybersecurity preparedness programs with a particular focus on combating ransomware. CISA also announced it would begin to use new administrative subpoena powers authorized under the 2021 National Defense Authorization Act to help it address ransomware attacks and other cyber threats [33]. Finally, the Justice Department created a new task force dedicated to rooting out and responding to the growing threat of ransomware [34]. This initiative demonstrates how one of the contenders found in this work is acting on the matter.

Russia, another example from our contender's list, is amid a public battle with the American White House because they imposed new sanctions on six Russian technology companies that provide support to the cyber program run by Putin's intelligence services linked to the hacking of the SolarWinds information technology company. The Homeland Security and Governmental Affairs Committee introduced bipartisan legislation to provide additional resources and better coordination for serious cyber attacks or any breaches that might risk the safety and security of Americans. The Ukrainian War is not mentioned here because it would need specific assessment of its own.

As for Brazil, the country seems to be taking slow steps into this new geopolitical reality. The Brazilian National Cyber Security Strategy (E-Ciber) addresses issues of cyber security of critical infrastructure and guides cyber defense [35]. Also, the National Defense Strategy (NDS) recognized the necessity to invest on Cyber Security that is independent from other nations in order to have an updated national defense system. Brazil should not be subject to foreign technology. However, both texts are silent on the matter of protection of commercial interests [35].

When it comes to cyber security, there is a knowledge gap between Brazilian capacity when compared to other countries. Episodes such as the Snowden case and the Stuxnet malware have brought notoriety to the matter [36]. It is relevant to point our that some of the countries publicized in those episodes are contenders to Brazilian Agribusiness [1], and there is nothing stopping them from using available cyber resources to benefit their industries in commercial negotiations [37].

In this setting, identification and protection of privileged data is fundamental to facing the market competitively. Thus, it is crucial to identify threats, threat actors and their capabilities, in order to prepare [38]. Cyber Threat Intelligence (CTI) has developed into a necessity so that agribusinesses are able to manage risk accordingly, become aware of vulnerabilities in time, and produce pertinent intelligence. Similar to other industries, protection of strategic data is a requirement for the preservation and expansion of Brazilian Agribusiness interests in the global market. Strategic actions to value Brazilian agribusiness must start with proper cyber security precautions.

*Context after Recent Cases*

Recent cases demonstrate that supply chains in general are not at a very high level of security, States and private companies need to invest heavily on dialogue and cyber risk management to specify minimum cyber security requirements for companies all according to CTI requisites Section 2.2.1. After all, a strategic approach to understand CTI might lead to sustainable long-term cyber security strategies.

States that hold a great dependency on their agricultural exports for internal revenue, such as Brazil, ought to reevaluate their definitions of critical infrastructure in order to embrace agribusinesses and supply chains, for a more up-to-date and commercial centered definition of Critical Infrastructure. Sector-specific rules should consider the national economic risks of disruption. Regulation for mandatory following of basic steps could begin a small revolution in the sector:

- Hire an Experienced Cyber team;
- Keep Security Software updated;
- Use Multi Step authentication;

- Teach Cyber Vigilance to employees.

On the other hand, the USA is sending a clear message regarding its growing interest in the sector. One might argue that the rise in supply chain attacks may be due in part to improved defenses against more rudimentary assaults, as it was seen in the USAHERDS, Section 2.4.3, and the John Deere, Section 2.4.2, cases.

Aside from regulation talk, countries might take other actions to target critical infrastructure cyber protection. Small and medium businesses need information and support to enter this high-tech environment with even a slim chance of securing themselves from attacks. Specifically, for the Brazilian beef agribusiness sector, this is a reality since many of our breeders fit such a category. Countries must also adopt Cybernetic Security as one of their political protection agenda points when negotiating with other nations. Attackers must be pressured and unveiled as wrongdoers. Finally, investigative teams must be attentive to choke points that aid attackers such as political shelter or financial outlines.

## 6. Conclusions

This paper proposed an assessment scheme that interfaces economic and political interests against potential cyber attack. In Section 2, we were able to introduce an analytics interface that targeted academic and corporate issues. In Section 2.2, the CTI matter was explained and connected to the agribusiness sector. We classified the main state contenders of Brazilian beef agribusiness, through the crossover of economic, political, and CTI evaluation. We believe that the intersection of CTI with economic and political components may lead to thorough and updated assessment for the unveiling of potential cyber threats.

It was possible to reach the goal of proposing a strategic assessment to introduce an analytics that target both academic and corporate issues based on CTI principles. Therefore, we were able to arrive at a list of the main state contenders of Brazilian meet agribusiness, as well as their cyber attack capacities, through economic, political, and CTI evaluation.

The main contribution of this study is described in Section 3, where we present a model to assess information to threat analysis of potential contenders. We believe such information is primordial to improve threat mitigation through the early diagnosis of potential threat actors while considering the creation of intelligence based on CTI principles. The Brazilian authorities may advance their planning on how to understand and classify the Food Sector within the scope of the Country's Cyber Security Legal Protection Framework. In Section 2, we discussed how such a business sector is already considered Critical Infrastructure in many other nations.

Despite this, legislative differences between CI definitions reflect national preferences, realities and needs. In addition to the core role that Brazilian agribusiness play world wide, one must wonder whether the Brazilian CI definition should be modernized to include agribusiness to obtain a fuller and up-to-date understanding of the cyber capabilities international scenario so that one may adapt appropriate defense mechanisms through mandatory security measures.

It is important to mention that the recent [39] article on the risk of fake controversies for Brazilian environmental policies was refuted by Embrapa. The Brazilian agriculture research company asserted that the Brazilian agribusiness has become the main focus of attacks nationally and internationally to create a negative image, reduce internal support, and disperse international consumers. The discussion is controversial considering the public dispute between academia and the heads of Embrapa, and it requires further investigation.

For future work, it is necessary to deepen the research on each contender and its history in attacks to fully understand what capabilities each holds, and how to properly analyse risk based on those capabilities, history, commercial context and intent, and with CTI principles at hand. On the other hand, we did not undertake a detailed and specific analysis about each country's cyber attack capacities; thus, future work may include further analysis on that front.

Also, competition is varied and fierce when it comes to agribusiness. Current commercial relations put Brazilian agricultural commodities in constant conflict for market and pricing to benefit its products. In 2019, Brazil was the largest exporter of beef in the world but, on the other hand, the average price of Brazilian product was among the cheapest when compared to 20 other exporters [1]. Such information alongside the findings in this work may lead to a deeper assessment on the influence front using cyber tools as well.

Finally, our results show that we are able to draw a list of threat actors that may act or have the potential to act in the cyber world against the Brazilian Beef Production Chain. Besides this, the analysis shows characteristics of each state regarding how they usually act in this space. In itself, this may support other relevant assessment for intelligence gathering on potential risks, actors, and actions.

**Author Contributions:** V.d.M.D.T. provided the creation of the assessment, developing the main concept of the analysis criteria. R.d.O.A. reviewed the methodology and cyber security considerations. R.R.N. reviewed the assessment and the methodology. G.A.M. reviewed the general considerations of economic aspects and possible impacts. All authors have read and agreed to the final version of the manuscript.

**Funding:** This work has the funding support of the ABIN grant 08/2019.

**Institutional Review Board Statement:** Not applicable.

**Informed Consent Statement:** Not applicable.

**Data Availability Statement:** Not applicable.

**Acknowledgments:** All the authors acknowledge the support of ABIN grant 08/2019. R.d.O.A. gratefully acknowledges the General Attorney of the Union—AGU grant 697.935/2019; the General Attorney's Office for the National Treasure—PGFN grant 23106.148934/2019-67. R.R.N. acknowledges the support from the UniAtenas University Center.

**Conflicts of Interest:** The authors declare no conflict of interest.

## Abbreviations

The following abbreviations are used in this manuscript

| | |
|---|---|
| KRITIS | Betreiber Kritischer Infrastrukturen |
| DP | Gross Domestic Product |
| OECD | Organisation for Economic Co-operation and Development |
| CTI | Cyber Threat intelligence |
| CI | Critical Infrastructure |
| CICARNE | Centro de Inteligência da Carne bovina |
| DOJ | The Justice Department |
| USA | United States of America |
| GSI/PR | Institutional Security Office of the Presidency of the Republic |
| CISA | Cyber security and Infrastructure Security |
| Embrapa | Empresa Brasileira de Pesquisa Agropecuária |
| ENISA | The European Union Agency for Cybersecutirty |
| EU | European Union |
| ISO | International Organization for Standardization ISO |
| ITC | International Trade Centre |
| ITU | International Telecommunication Union |
| IT | Information Technology |
| NCPI | National Cyber Power Index |
| NEI | National Economic Intelligence |
| OSINT | open source intelligence |
| Saas | Software-as-a-Service |
| TISN | Trusted Information Sharing Network |
| UK | United Kingdom |

USA          United States of America

USAHerds      Animal Health Emergency Reporting Diagnostic System

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
