# Peer review of "Strategic Assessment of Cyber Security Contenders to the Brazilian Agribusiness in the Beef Sector"

_information, doi:10.3390/info13090431_

Round 1

Reviewer 1 Report

The manuscript addresses the Brazilian meet agribusiness from the perspective of cyber Threat intelligence (CTI) with a strategic assessment based on an analytical scheme. It attempts to describe and classify contenders (nation-states) along with their offensive cyber capabilities, producing a model to assess and mitigate potential cyber attack threats.

The manuscript is well structured and the presentation is clear and organized with relevant and adequate references. Methodologically, the paper uses a bottom-up approach to reach its conclusions, and sometimes the overall line of argumentation is not clear, which makes it somewhat hard to follow. Sections seem to be juxtaposed rather than chained in an argument. Nevertheless, the manuscript is consistent (no inconsistencies detected) and the conclusions are justified. Since the paper is more argumentative and qualitative, the whole argument must be reviewed to clarify the line of reasoning and explicit the relations between sections.

The paper starts by describing the current and forecast pressure on the food market and places the food industry as part of the critical infrastructure showing how recent technological innovation makes it prone to cyber-attacks and hybrid warfare. CTI and recent attack cases are presented and discussed (Sec. 3). It should be noted that of the four presented cases, only one is related to agribusiness (Sec. 3.2).

There is almost no correlation or comparison between this case and the others. The section on advancements from cases (Sec. 3.5) deals mostly uniquely with the agribusiness case (3.2). These unsatisfactory points must be corrected.

Sec. 4 brings CTI into the discussion, but, in terms of building an argument, its rationale with the previous section is not clear and it also seems isolated. Another point is that the section title seems misleading: the section does not deal with capability but with its estimation. Sec. 4.2 also needs some adjustments in text and title.

Sec. 5 is consistent in presenting the adoption of technology and its impact, mapping with recent specific incidents. Perhaps it should be placed close to the discussion of incidents.

Sec. 6 is where the main contribution is expected. It successfully analyses and identifies contenders in the beef market. However, the way it, there is not much related to the cyber threats and CTI itself, these topics must be enhanced in the text.

The conclusion states; "The proposal of this paper is for an assessment scheme that interfaces economic and political interests against cyber attack capabilities". However, this contrast with cyber attack capabilities needs to be made more evident and clear.

As a final remark, the overall line argumentation must be improved.

Author Response

Journal: Information

Original Manuscript ID: information-1834550

Original Article Title: “Strategic Assessment of Cyber Security Contenders to the Brazilian Agribusiness in the Beef Sector”

To: Assigned Editor, Lily Yang

Re: Response to reviewers

Brasília, August 7th , 2022.

Dear Editor,

Firstly, we would like to express our sincere gratitude and appreciation to you and the reviewers for volunteering their time in reviewing our paper and providing us with valuable comments. We also thank you for allowing us to resubmit the new version of our manuscript after the corrections we have made to the text.

Please find below a detailed point-by-point response (italicized text) to each of the reviewers' comments (boldface text). Also, we send two versions of the manuscript, the first version with all revised points in yellow highlight, and the second version of the manuscript without highlights in PDF format. Finally, we indicate the line in which the changes may be found in the new and revised manuscript.

We agree the new version of the manuscript has improved due to the valuable comments of the reviewers. We hope that with such improvements and clearness, the revised manuscript addresses satisfactorily all the raised concerns.

We look forward to hear from you.

Yours Sincerely,

The Authors.

Response to Reviewer 1 Comments

The manuscript addresses the Brazilian meet agribusiness from the perspective of cyber Threat intelligence (CTI) with a strategic assessment based on an analytical scheme. It attempts to describe and classify contenders (nation-states) along with their offensive cyber capabilities, producing a model to assess and mitigate potential cyber attack threats.

1) The manuscript is well structured and the presentation is clear and organized with relevant and adequate references. Methodologically, the paper uses a bottom-up approach to reach its conclusions, and sometimes the overall line of argumentation is not clear, which makes it somewhat hard to follow. Sections seem to be juxtaposed rather than chained in an argument. Nevertheless, the manuscript is consistent (no inconsistencies detected) and the conclusions are justified. Since the paper is more argumentative and qualitative, the whole argument must be reviewed to clarify the line of reasoning and explicit the relations between sections.

We are very grateful for the reviewer satisfaction with our work. We performed a general review of the approach to follow IMRAD principles to make the line of argumentation clearer and the reading more fluent. Sections names and order were revised and we believe that clarified the line of reasoning. Additionally, we connected the sections with coherent connection sentences and we added intra text references, we are confident those changes will explicit the relations between the sections and improve overall quality of the text.

2) The paper starts by describing the current and forecast pressure on the food market and places the food industry as part of the critical infrastructure showing how recent technological innovation makes it prone to cyber-attacks and hybrid warfare. CTI and recent attack cases are presented and discussed (Sec. 3). It should be noted that of the four presented cases, only one is related to agribusiness (Sec. 3.2).

We thank the reviewer for this observation. In response to the reviewer observation, we moved the cases to the related works section and we added an explanation as to why those cases are relevant (see lines 353-360). Additionally, we presented the point of why it is important to consider critical supply chains as a whole in the case analysis (see lines 84-87 and 183-187). We also described the importance of this in the Introduction and in the Discussions section (see lines 41-42, 54-57, and 625-629). We highlight that the entire work`s section organization and order was changed

3) There is almost no correlation or comparison between this case and the others. The section on advancements from cases (Sec. 3.5) deals mostly uniquely with the agribusiness case (3.2). These unsatisfactory points must be corrected.

We would like to thank the reviewer for this. In attention of the observation made by the reviewer about the lack of correlation between the presented cases, we have changed the text to present the reasons why it is important to take into account attack cases against supply chains as well (see lines 41-42, 54-57, and 625-629). Additionally, we moved the cases section to the related works section in order to make a straightforward point as how this information is relevant background knowledge to fully understand the work’s line of thought (see new section 2.4, line 252-401).

4) Sec. 4 brings CTI into the discussion, but, in terms of building an argument, its rationale with the previous section is not clear and it also seems isolated. Another point is that the section title seems misleading: the section does not deal with capability but with its estimation. Sec. 4.2 also needs some adjustments in text and title.

We thank the reviewer for this important remark. In order to clear the relevance of CTI to the work’s line of thought, we moved this topic to the Related Works section (see section 2.2.1, lines 244-284, and section 2.3, 285-351), and corelated it to the previous section about TI (see lines 235-237). The section title was changed to bring a more factual message to the reader (see line 285). Section 4.2 of the first manuscript was reevaluated and its content was reviewed and placed in the discussions section to assess countries response and capacities in the situation of attack (see lines 568-615).

5) Sec. 5 is consistent in presenting the adoption of technology and its impact, mapping with recent specific incidents. Perhaps it should be placed close to the discussion of incidents.

In attention to the reviewer point, we moved section 5 and incidents section to the Related works and background section to make it clear that those sections are part of the background information needed to fully understand the work’s line of thought following IMRAD’s principles (see sections 2.1.1 and 2.1.2, lines 158-220).

6) Sec. 6 is where the main contribution is expected. It successfully analyses and identifies contenders in the beef market. However, the way it, there is not much related to the cyber threats and CTI itself, these topics must be enhanced in the text.

We would like to thank the reviewer for this observation. In response,  we emphasize that we placed the CTI and TI topics as related works (see sections 2.2 and 2.2.1,  lines 220-284), where we explain to the reader the relevance of CTI principles to adequately grasp the results of the work. Also, the topics were reviewed and enhanced in the text (see lines 497, 506, 540, 581, 618, 628, and 654-658).

7) The conclusion states; "The proposal of this paper is for an assessment scheme that interfaces economic and political interests against cyber attack capabilities". However, this contrast with cyber attack capabilities needs to be made more evident and clear.

We appreciate the evaluation and the point well raised, we improved the explanation of the assessment by adding a Proposed Methods and Criteria and a Discussion section (see sections 3 and 5, lines 402-451, and 521-651) . We believe that those changes made the contrast with CTI clearer.

8) As a final remark, the overall line argumentation must be improved.

We appreciate the reviewer comment. We would like to inform that the text was fully reviewed, reevaluated and reorganized to reach a more agreeable line of argumentation to the reader. We added a section on Proposed Methods and Criteria Evaluation, a section on Results and Evaluations, and one on Discussions see sections 3, 4 and 5, lines 402-451, 453-520 and 521-651). Additionally, we made the necessary changes in the writing to guarantee text cohesion and coherence.

Reviewer 2 Report

The article suggests a current and attractive topic for the academy. The effort is evident, but it requires adjustments for better understanding and quality.

The study on this topic is fascinating. The structure is clear and logical and challenging. The research is timely and worthwhile.

Authors should follow the style of a structured abstract based on the IMRAD structure of a paper. The abstract should briefly state the purpose of the research, the principal results, and significant conclusions. An abstract is often presented separately from the article, so it must be able to stand alone.

I hope you find the following observations helpful:

Materials and methods: I found this section very important for the paper's readability. Methods should be described in detail. I think the research procedure could be much more clearly described using a diagram, highlighting its potential and limit.

The references, although varied, need up-to-date. It says a lot.

Authors should consider more previous works (e.g., theoretical, conceptual, and empirical reviews) published in the literature. Authors should discuss the results and how they can be interpreted from the perspective of previously published studies. I suggest adding to the references list the following papers

·         Hu Z., Khokhlachova Yu., Sydorenko V., Opirskyy I. Method for Optimization of Information Security Systems Behavior under Conditions of Influences. International Journal of Intelligent Systems and Applications (IJISA), Vol.9, No.12, 46-58 (2017). DOI: 10.5815/ijisa.2017.12.05

·         Hu Z., Odarchenko R., Gnatyuk S., Zaliskyi M., Chaplits A., Bondar S., Borovik V. Statistical Techniques for Detecting Cyberattacks on Computer Networks Based on an Analysis of Abnormal Traffic Behavior. International Journal of Computer Network and Information Security (IJCNIS). Vol.12, No.6, pp.1-13, 2020. DOI: 10.5815/ijcnis.2020.06.01

I strongly recommend adding these works to the list of references.

Authors should discuss the results and how they can be interpreted from the perspective of previously published studies.

The diagrammatic presentation of the study research will be the most substantial section of this work. I suggest adding a visual presentation of obtained outcomes in section Results.

Structure: Articles should be reformatted according to a standard structure, which is set out in the instructions for the authors of the Journal (sections are Introduction, Materials and Methods, Results, and Discussions, Conclusion). See template.

I also suggest a grammar and spelling review. 

The conclusion is thorough.

Author Response

Journal: Information

Original Manuscript ID: information-1834550

Original Article Title: “Strategic Assessment of Cyber Security Contenders to the Brazilian Agribusiness in the Beef Sector”

To: Assigned Editor, Lily Yang

Re: Response to reviewers

Brasília, August 7th , 2022.

Dear Editor,

Firstly, we would like to express our sincere gratitude and appreciation to you and the reviewers for volunteering their time in reviewing our paper and providing us with valuable comments. We also thank you for allowing us to resubmit the new version of our manuscript after the corrections we have made to the text.

Please find below a detailed point-by-point response (italicized text) to each of the reviewers' comments (boldface text). Also, we send two versions of the manuscript, the first version with all revised points in yellow highlight, and the second version of the manuscript without highlights in PDF format. Finally, we indicate the line in which the changes may be found in the new and revised manuscript.

We agree the new version of the manuscript has improved due to the valuable comments of the reviewers. We hope that with such improvements and clearness, the revised manuscript addresses satisfactorily all the raised concerns.

We look forward to hear from you.

Yours Sincerely,

The Authors.

Response to Reviewer 2 Comments

1)Structure: Articles should be reformatted according to a standard structure, which is set out in the instructions for the authors of the Journal (sections are Introduction, Materials and Methods, Results, and Discussions, Conclusion). See template.

The article suggests a current and attractive topic for the academy. The effort is evident, but it requires adjustments for better understanding and quality.

We are very grateful for the reviewer satisfaction with our work. We performed a general review of the approach to follow IMRAD principles to make the line of argumentation clearer and the reading more fluent. Sections names and order were revised and we believe that clarified the line of reasoning. Additionally, we connected the sections with coherent connection sentences and we added intra text references, we are confident those changes will explicit the relations between the sections and improve overall quality of the text.

2) The study on this topic is fascinating. The structure is clear and logical and challenging. The research is timely and worthwhile.

We appreciate the reviewer comment. Thank you. We are confident the review of the text has improved its quality.

3) Authors should follow the style of a structured abstract based on the IMRAD structure of a paper. The abstract should briefly state the purpose of the research, the principal results, and significant conclusions. An abstract is often presented separately from the article, so it must be able to stand alone.

In attention to the reviewer request, we changed the abstract structure based on IMRAD principles, and it now states the purpose of the research, the principal results, and significant conclusions. Additionally, the entire structure of the paper was revised to fit those guidelines from IMRAD, we added a section on Proposed Methods and Criteria Evaluation, a section on Results and Evaluations, and one on Discussions see sections 3, 4 and 5, lines 402-451, 453-520 and 521-651). Finally, we made the necessary changes in writing to guarantee text cohesion and coherence. 

4) Materials and methods: I found this section very important for the paper's readability. Methods should be described in detail. I think the research procedure could be much more clearly described using a diagram, highlighting its potential and limit.

We thank the reviewer for this observation. We have added a specific and detailed section called Proposed Methods and Criteria Evaluation (see section 3, lines , lines 402-451). The research procedure was described in a diagram as suggested (see figure 1 between lines 441 and 442). Finally, the method potential and limit are highlighted in the section Contributions and Limitations of this Work (see section 1.1, lines 47-60) .

5) The references, although varied, need up-to-date. It says a lot. Authors should consider more previous works (e.g., theoretical, conceptual, and empirical reviews) published in the literature. Authors should discuss the results and how they can be interpreted from the perspective of previously published studies. I suggest adding to the references list the following papers:

  • Hu Z., Khokhlachova Yu., Sydorenko V., Opirskyy I. Method for Optimization of Information Security Systems Behavior under Conditions of Influences. International Journal of Intelligent Systems and Applications (IJISA), Vol.9, No.12, 46-58 (2017). DOI: 10.5815/ijisa.2017.12.05
  • Hu Z., Odarchenko R., Gnatyuk S., Zaliskyi M., Chaplits A., Bondar S., Borovik V. Statistical Techniques for Detecting Cyberattacks on Computer Networks Based on an Analysis of Abnormal Traffic Behavior. International Journal of Computer Network and Information Security (IJCNIS). Vol.12, No.6, pp.1-13, 2020. DOI: 10.5815/ijcnis.2020.06.01

I strongly recommend adding these works to the list of references.

We appreciate the reviewer’s observation. And we have reviewed and cited the mentioned papers accordingly (see line 248).

6) Authors should discuss the results and how they can be interpreted from the perspective of previously published studies.

We thank the reviewer’s concern. We have added a section on discussions where we were able to interpret the findings from the perspective of previously published studies following IMRAD`s principles (see section 5 lines 521-651).

7) The diagrammatic presentation of the study research will be the most substantial section of this work. I suggest adding a visual presentation of obtained outcomes in section Results.

We thank the reviewer observation. We created a section called results and evaluations where we added a step-by-step visual presentation of the presented scheme and a complete presentation of the outcome, see tables 1 - 5. All to give an unambiguous display of the study research.

8) I also suggest a grammar and spelling review.

We thank the reviewer for this observation. We performed a grammatical and spelling review of the entire manuscript as oriented.

Round 2

Reviewer 1 Report

The authors have extensively improved the manuscript. All the issues addressed in the review were considered properly and to my satisfaction.